# Microbial Transformation of the Sesquiterpene Lactone, Vulgarin, by *Aspergillus niger*

**DOI:** 10.3390/molecules28093729

**Published:** 2023-04-26

**Authors:** Reem A. ElGamal, Amal A. Galala, Maged S. Abdel-Kader, Farid A. Badria, Amal F. Soliman

**Affiliations:** 1Pharmacognosy Department, Faculty of Pharmacy, Mansoura University, El Mansoura 35516, Egypt; reem.a.elgamal@hotmail.com (R.A.E.); amal_galala@yahoo.com (A.A.G.); amalsoliman134@mans.edu.eg (A.F.S.); 2Department of Pharmacognosy, College of Pharmacy, Prince Sattam Bin Abdulaziz University, Al-Kharj 11942, Saudi Arabia; m.youssef@psau.edu.sa; 3Department of Pharmacognosy, Faculty of Pharmacy, Alexandria University, Alexandria 21215, Egypt

**Keywords:** vulgarin, biotransformation, *Aspergillus niger*, vulgarin metabolites, COX inhibitors

## Abstract

The biotransformation of vulgarin **(1)**, an eudesmanolides-type sesquiterpene lactone obtained from *Artemisia judaica*, by the microorganism, *Aspergillus niger*, was carried out to give three more polar metabolites; 1-*epi*-tetrahydrovulgarin (1α,4α-dihydroxy-5αH,6,11βH-eudesman-6,12-olide **(2)**, 20% yield, 1α,4α-dihydroxyeudesm-2-en-5αH,6,11βH-6,12-olide **(3a)**, 10% yield, and C-1 epimeric mixture **(3a, b)**, 4% yield, in a ratio of 4:1, **3a/3b**. The structures of vulgarin and its metabolites were elucidated by 1 and 2D NMR spectroscopy in conjunction with HRESIMS. Metabolites **(3a)** and **(3b)** are epimers, and they are reported here for the first time as new metabolites obtained by biotransformation by selective reduction at C-1. Vulgarin and its metabolites were evaluated as anti-inflammatory agents using the human cyclooxygenase (COX) inhibitory assay. The obtained data showed that **(1)** exhibited a good preferential inhibitory activity towards COX-2 (IC50 = 07.21 ± 0.10) and had a moderate effect on COX-1 (IC50 = 11.32 ± 0.24). Meanwhile, its metabolite **(3a)** retained a selective inhibitory activity against COX-1 (IC50 = 15.70 ± 0.51). In conclusion, the results of this study revealed the necessity of the presence α, β unsaturated carbonyl group in **(1)** for better COX-2 inhibitory activity. On the other hand, the selectivity of **(1)** as COX-1 inhibitor may be enhanced via the reduction of C-1 carbonyl group.

## 1. Introduction

Microbial transformation is the specific modification of a definite compound to a distinct product with structural similarity with the use of biological catalysts including microorganisms, mainly fungi. It is an alternative to chemical reactions for searching for new derivatives with enhanced biological activities [1]. In recent years, using the frequency of the microbial transformation technique has increasing significantly, from there being a limited number of trials to it being highly active area in green chemistry, including the preparation of pharmaceutical products. Microbes now are being used as biological shelf reagents [2,3].

One of the main advantages of microbial transformation is the selective introduction of functional groups to the carbon skeleton of xenobiotic resulting for the production of new metabolites, which are difficult to be obtained by chemical reactions. Moreover, microbial transformation helps to study the metabolic fate of the xenobiotics, as it is suggested that microbial transformation can mimic the mammalian metabolism. Therefore, microbial transformation can be an alternative to animal models to study xenobiotic metabolism [2]. The main goal of biotransformation is the conversion of poorly excreted lipophilic molecules into more easily excreted hydrophilic metabolites [4].

*Aspergillus niger* is a filamentous fungus and one of the most common species of the genus *Aspergillus*, and it is found in soil, decaying vegetation, seeds, and grains. It is one of the most important microorganisms used in biotechnology. *A. niger* is used for biotransformation and waste treatment. Since the 1960s, *A. niger* has been a source of a variety of enzymes that are well developed as technical aids in fruit processing and baking. In addition to its industrial uses in the production of citric acid and extracellular enzymes. *A. niger* has the ability to produce not only proteins and enzymes at high concentrations, but also pharmaceuticals that are beneficial for humans and animals. Intense research on *A. niger* over the last decade has resulted in a range of new products and processes [5].

Eudesmanolides-type sesquiterpene lactone, vulgarin **(1)**, was isolated from different *Artemisia* species (family Asteraceae), including *Artemisia vulgaris*, after which it was named [6]. Vulgarin was assigned other names such as judaicin from *Artemisia judaica* [7]. In addition, vulgarin can be obtained by microbial transformation [8], as well as chemical the reduction of peroxyvulgarin [9]. Vulgarin **(1)** is a cytotoxic agent due to the presence of α, β unsaturated ketone [10]. It was found to be a promising candidate for treatment of different illness because of its potent anti-inflammatory, hypoglycemic, anti-bacterial, and anti-tumor activities [11,12,13,14]. The multiple biological activities of vulgarin make it an attractive target for microbial transformation studies.

Vulgarin was previously reported to undergo microbial transformation, and three metabolites were obtained by two different microorganisms. *Beauveria bassiana* (ATCC 7159) converted vulgarin to one metabolite; 1α,4α-dihydroxy-5αH,6,11βH-eudesman-6,12-olide (1-*epi*-tetrahydrovulgarin), while the yeast *Hansenula anomala* (ATCC 20170) yielded two metabolites; 4 α -hydroxy-1-oxo-5 α H,6,11βH-eudesman-6,12-olide (dihydrovulgarin) and 3 α,4 α -dihydroxy-1-oxo-5 α H,6,11βH-eudesman-6,12-olide (3 α-hydroxy dihydrovulgarin) [15].

The prolonged use of non-selective non-steroidal anti-inflammatory drugs (NSAIDs) results in severe side effects such as gastrointestinal hemorrhage due to inhibition of cyclooxygenase-1 (COX-1) enzyme [16], while most of the COX-2 selective drugs have been found to cause cardiovascular problems [17]. Consequently, there is a strong need to look for anti-inflammatory agents of a natural origin with minimum side effects.

The objective of this work is to utilize microorganisms for reinvestigating the biotransformation of vulgarin for the production of metabolites with enhanced anti-inflammatory activity and to study the metabolism of vulgarin by liver enzymes when used as an anti-inflammatory drug depending on the capability of microbial transformation to mimic mammalian metabolism. Moreover, this work aimed to study enzymatic reactions carried out by microorganisms, which have the advantage of selectivity over chemical reactions.

## 2. Results and Discussion

### 2.1. Structure Elucidation of Vulgarin Metabolites

The preparative incubation of vulgarin with *Aspergillus niger* (ATCC 10549) was carried out. *A. niger* metabolized vulgarin **(1)** after 12 days incubation into three metabolites (**2)**, (**3a),** and (**3a, b)** (Figure 1), which are more polar than **(1)** is. After the chromatographic isolation and purification of three vulgarin metabolites, the identification of isolated pure metabolites was achieved using different spectroscopic techniques, including 1D and 2D NMR and MS analyses (Appendix A).

#### 2.1.1. Structural Elucidation of Metabolite **(2)**

NMR spectra for vulgarin and the three metabolites showed a close resemblance with small differences, as shown in Table 1 and Table 2. Metabolite **(2)** showed the absence of α, β unsaturated carbonyl; instead it showed one more oxygenated methine at ⸹C 73.3 ppm and two extra methylenes at 33.0 and 36.4 ppm. HRESIMS showed a quasi-molecular ion peak at 291.1564 (M+Na)^+^ calculated for C_15_H_24_O_4_ (291.1572) with four degrees of unsaturation. As per Table 1, ^1^HNMR revealed the presence of three methyls at ⸹H 1.00 (S), 1.23(d, *J* = 6.9 Hz), and 1.36 (S), assigned to methyls 14, 13, and 15, respectively. A downfield double of doublet lactonic proton resonating at ⸹H 4.07 (dd, *J* = 10.7, 10.6 Hz) assigned to H6 and a broad singlet resonating at ⸹H 3.41 assigned to H1 are the last protons connecting directly to their corresponding carbon via one bond length in HSQC experiment at ⸹c 81.4 and 73.3, respectively. The above data, together with ^13^CNMR, confirmed the structure of **(2)** to be vulgarin metabolite with the replacement of carbonyl at C1 by oxygenated methine group and the saturation of the olefenic double bond between carbons 2 and 3. Moreover, an APT experiment showed a total of 15 carbon atoms distinguished into three methyles, 4 methylene, 5 methins, and three quaternaries. The three quaternaries resonated at 41.6, 71.5, and 178.7, assigned to C10, C4 and C12, respectively. The key connectivity between protons 1 and 2; 2 and 3; 5 and 6; 7 and 6, 8,11; 8 and 7,9 were proved by the observed cross peaks correlation between these protons in the COSY experiment.

An NOESY experiment was important to determine the orientation of the hydroxyl group at C1 as α oriented by observation of NOESY cross-speak correlations between H1 and Me14 and Me15 (Figure 2). On the other hand, the α orientation of H5 was proved by the lack of NOE correlation with Me15, while the orientations at C6, C7, and C11 were proved to be as the same of those of vulgarin. The structure of compound **(2)** was verified as 1-*epi* tetrahydrovulgarin (20% yield without optimization), previously isolated from *Artemisia canariensis* [18] and obtained as metabolite by fungus *Beauveria bassiana* transformation [15].

#### 2.1.2. Structural Elucidation of Metabolite **(3a)**

Such as **(2)**, compound **(3a)** showed similarity to vulgarin, except for the lack of carbonyl at C1 (^13^CNMR), and instead, an extra 2ry alcoholic group appeared in ^1^HNMR at ⸹H 3.49 (br*s*) connected to its carbon in the HSQC experiment at ⸹C 71.3. As per Table 1, ^1^HNMR showed, in addition to oxygenated methine at 3.49 ppm, an olfenic double bond at ⸹H 5.94 (dd, *J* = 10.1,5.6) and 5.70 (d, *J* = 10.1 Hz) assigned to a proton at C2 and 3. The proton at C2 was coupled to that one at C1 and C3 in ^1^H-^1^HCOSY experiment, which give an idea about the replacement of carbonyl at C1 in vulgarin by a secondary alcoholic group. Additionally, ^1^HNMR showed three methyls, two of them are singlets resonating at ⸹H 0.97 and 1.39, and the third one is doublet, and it appeared at ⸹H 1.25 (d, *J* = 6.9 Hz), assigned to methyls 14, 15, and 13, respectively. A lactonic proton appeared at ⸹H 4.05 and was split as dd (*J* = 10.7 and 11.4 Hz) and assigned to proton at C6 and connected to its direct attached carbon at ⸹c 81.5 in 2D HSQC experiments. The remaining protons were in close similarity to those of vulgarin, and this confirming the eudesamnolide basic nucleus for **(3a)**.

The DEPT experiment and ^13^CNMR revealed a total of 15 carbon atoms with olefinic double bond carbons at ⸹c 127.3 and 136.2 lactonic carbon (cyclic ester) at ⸹c 179.0, assigned to C2, C3, and C12, respectively. Additionally, an oxygenated quaternary carbon appeared at ⸹c 70.4 and was assigned to C4. The final structure of metabolites **(3a)** was proved to be 1α,4α-dihydroxyeudesm-2-en-5α,6β,11β-6,12-olide **(3a)** (15% yield without optimization) based on the significant *J*_2_ and *J*_3_ HMBC correlations from H1 to C2, C3, C5, and C10; from H2 to C3, C5 and C10; from methyl 13 to C7; C11 and C12. The location of OH at C1 was confirmed on the lower surface of the molecule (α oriented) by noticing the NOESY cross-peaks between H1 and methyl 14 and methyl 15 (Figure 2). The aforementioned data confirmed the structure of **(3)** to be 1α,4α-dihydroxyeudesm-2-en-5α,6β,11β-6,12-olide **(3)** (10% yield without optimization) previously isolated from the aerial parts of *Artemsia spicigera* [19] and reported here for the first time by microbial transformation.

#### 2.1.3. Structural Elucidation of Metabolite **(3a, b)**

Metabolite **(3a, b)** (4% yield without optimization) was isolated as a mixture of two compounds. It showed two molecular ion peaks, one at M+ Na at 289.2706 corresponding to **(3a)**, and the other for **(3b)** appeared at 289.1405 for M-2 in HRESIMS. NMR spectra showed a little difference between **(3a)** and **(3b),** suggesting that the two compounds are isomers of each other. The main differences were noticed in ^13^CNMR (Table 2) for C1, C5, C9, and C10, with downfield shifts of ⸹H 2.1, 1.8, 1.0, and 1.4 ppm, respectively. A significant difference was observed for C1 and C5 chemical shifts. Moreover, ^1^HNMR for oxygenated carbon at position 1 shifted upfield from 3.49 in **(3a)** to 3.41 ppm in **(3b)**. The methyl chemical shift at position 14 was also downfield shifted from ⸹H 0.97 in **(3a)** to 1.00 ppm in **(3b),** while for methyls 13 and 14, they showed a little upfield shift from 1.25 to 1.22 ppm in **(3a)** and from 1.39 **(3a)** to 1.36 **(3b)**, respectively. The rest of NMR spectral data were almost the same for both metabolites, as shown in Table 1 and Table 2.

The NOESY experiment was valuable for confirming the orientation of OH at position 1 to be β orientation by the absence of NOESY cross-peaks between H1 and methyls 14 and 15 (Figure 2). The above-mentioned data confirmed that **(3a)** and **(3b)** are epimers of each other and present in the mixture at a ratio of 4:1 **3a**/**3b**. Metabolites **(3a)** and **(3b)** are reported here for the first time as new metabolites obtained by the biotransformation of vulgarin, and the structure of **(3b)** is 1β,4α- dihydroxyeudesm-2-en-5α,6β,11β-6,12-olide.

**Figure 2 molecules-28-03729-f002:**
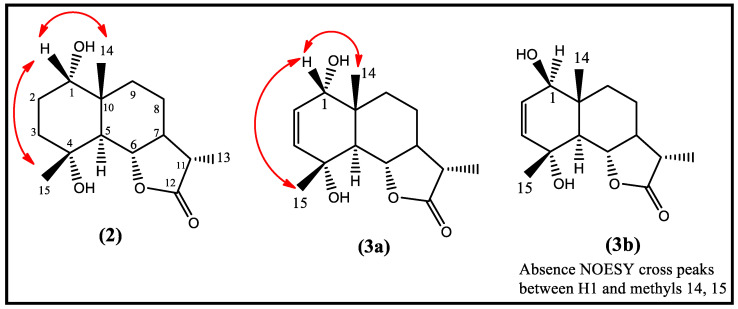
Significant NOESY correlations of vulgarin metabolites.

### 2.2. Proposed Mechanism of Selective Reduction

*Aspergillus niger* is considered as a source of oxidoreductases enzymes. NAD(P)H-dependent oxidoreductases enzymes, such as alcohol dehydrogenase (ADH), can selectively control the locus of reduction independent from multiple functional groups present within the starting compound [20]. Figure 3 showed the chemoselective reduction of the C-1 carbonyl group of vulgarin in the presence of a C2=C3 double bond. Moreover, the selective reduction of the Δ^2,3^ double bond producing dihydrovulgarin has been previously reported [15].

### 2.3. COX-Inhibitory Activity of Vulgarin and Its Metabolites

Interestingly, a selective COX-1 inhibitor was equipotent to COX-2 selective inhibitor (celecoxib) for inhibiting PG formation in an inflammatory exudate. These findings and other data propose that the functions of COX-1 and COX-2 might be more complicated than they have been initially estimated and that COX-1 inhibition may contribute to the inhibition of PG production in inflammatory exudates. Hence, a combined inhibition of COX-1 and COX-2 may lead to the more efficient inhibition of chronic inflammation as compared to the selective inhibition of COX-1 or COX-2 [21].

The anti-inflammatory activity of vulgarin and its metabolites was evaluated by measuring their ability to inhibit COX-1 and COX-2 enzymes and comparing their inhibitory activity with that of a reference compounds, celecoxib and indomethacin (Table 3) [22,23]. Based upon the % of inhibitory activity of tested compounds on COX-1 and COX-2, IC_50_ (µM/mL) were determined as shown in Table 4.

The results (Table 3) showed that vulgarin showed a good preferential inhibitory activity towards COX-2 and had a moderate effect on COX-1 (% of inhibition of COX-1 = 45.61%, COX-2 = 60.69%). Moreover, other compounds; **(2)**, **(3a)**, and **(3a, b epimeric mixture)** showed different % of inhibition of (COX-1, COX-2); (9.43%, 6.80%), (53.60%, 32.7%), and (48.40%, 26.17%), respectively. Metabolite **(2)** showed a weak inhibitory effect on COX-1 and COX-2 (Table 3). Metabolites **(3a)** and **(3a, b)** exhibited a selective inhibitory activity against COX-1 and had a weak effect on COX-2.

Figure 4 showed that **(1)** exhibited a good preferential inhibitory activity towards COX-2 (IC50 = 07.21 ± 0.10) and had a moderate effect on COX-1 (IC50 = 11.32 ± 0.24). Meanwhile, its metabolite **(2)** lost the inhibitory activity on both enzymes and metabolite **(3a)** retained a selective inhibitory activity against COX-1 (IC50 = 15.70 ± 0.51). In conclusion, the results of this study reveal the necessity of the presence α, β unsaturated carbonyl group in **(1)** for better COX-2 inhibitory activity. On the other hand, the selectivity of **(1)** as COX-1 inhibitor may be enhanced via the reduction of the C-1 carbonyl group.

## 3. Materials and Methods

### 3.1. General

Using CDCl_3_ solvent and TMS as internal standard for chemical shifts, ^1^H-NMR and ^13^C-NMR spectra were recorded on Bruker DRX 500 and 700 NMR spectrometer operating at 500 and 700 MHz and 125 and 175 MHz, respectively. Chemical shifts (δ) are expressed in ppm with reference to TMS resonance. For ^13^C NMR spectra, the number of attached protons was determined by DEPT 135° and 2D NMR data were obtained using the standard pulse sequence of the Bruker DRX-500 for COSY, HMQC, HMBC, and NOESY. The accurate mass determination was achieved with a JEOL JMS-700 High-Resolution Mass Spectrophotometer (JEOL USA Inc., Peabody, MA, USA) in positive and negative modes.

Normal phase chromatography was carried out using silica gel 60–120 mesh (Alpha Chemika, Mumbai, India) packed by the wet method in the specific solvents. Media ingredients and solvents used for extraction and chromatographic separation were purchased from El-Nasr Company for Pharmaceutical Chemicals, Egypt. Analytical thin layer chromatography was performed on precoated silica gel 60 GF_254_ on aluminum sheets (Merck, Germany). Plates were developed in different solvent mixtures, and developed chromatograms were visualized under UV light 254 and 366, and spots were made visible by spraying with vanillin sulphoric spray reagent after warming in an oven preheated to 105 °C for 1 min.

### 3.2. Plant Material

*Artemisia judaica L.* plant was collected from the Huraymila region, to the west of Riyadh city, in February 2019. The plant was identified by Dr. Mohammad Atiqur Rahman, a taxonomist of MAP-PRC at the Faculty of Pharmacy, King Saud University, Riyadh, Saudi Arabia. A voucher specimen (#16723) was kept in the herbarium in this center.

### 3.3. Extraction and Isolation of Vulgarin

The dried ground aerial parts (3 kg) of *A. judaica* were extracted by 95% ethanol (30 L) until exhaustion. The extract was evaporated under vacuum using rotatory evaporator, leaving a dark green residue which was subjected to liquid–liquid fractionated using solvents; petroleum ether, chloroform, and ethylacetate, respectively. The chloroform fraction (30 g) was chromatographed on a Sephadex LH20 column (100 × 6 cm i.d., 600 g) using gradient elution with solvent system (chloroform/acetone 75:25), followed by a gradual increase in acetone by 25% until it reached 100% acetone; then, a gradient of acetone/methanol was used by increasing methanol by 25% until it reached 100% methanol. Fractions of 100 mL were collected, and similar fractions were pooled together to give 5 major fractions based on their TLC profile. The fraction eluted with chloroform/acetone 25:75 (8 g) was purified using a silica gel column (80 × 3.5 cm i.d., 200 g) using chloroform/methanol solvent system in a gradient elution. The fractions eluted with solvent system (chloroform/methanol 95:5) provided 2.4 g of vulgarin after repeated crystallization with methanol.

### 3.4. Microorganisms

Microorganisms were obtained from ATCC (American Type Culture Collection), NRRL (Northern Regional Research Laboratories), and NBRC (Biological Resource Centre, NITE, Tokyo, Japan) and stored at 4 °C in PDA (Potato Dextrose Agar) medium. Screening procedures to transform vulgarin, were carried out on the following 20 microorganisms: *Aspergilus niger* (ATCC 10549), *Aspergilus flavipes* (ATCC 11013), *Aspergilus versicolor* (AUMC 150201), *Cordyceps (Isaria) sinclairii* (ATCC 24400), *Cordyceps sinensis* (NBRC 610453), *Rhodotorula rubra* (NRRL y1592), *Rhizopus species* (ATCC 36060), *Gymnascella citrina* (NRRL 6050), *Alternaria alternata* (AUMC 150207), *Penicillium chryzogenium* (ATCC 9490) and *Rhizopogen sp.* (ATCC 6060), *Cunninghamella blackesleeana* (NRRL1369), *Phlebia firma* (ATCC 64378), *Phlebia segregate* (ATCC 90073), *Mucor species* (AUMC 150209), *Penicillium chrysogeneum* (ATCC 9480), *Ceriporia spissa* (ATCC62024), *Saccharomyces cerivisae* (Lyophilized yeast cell), and two strains of *Cunninghamella elegans—Cunninghamella elegans* (NRRL 2310) and *Cunninghamella elegans* (NRRL 1392).

*Aspergilus niger* (ATCC 10549) was selected for preparative scale fermentation, as it reproducibly produced metabolites in higher yield.

### 3.5. Screening Procedures

The screening process was carried out by adapting two stages fermentation protocol [24] in a liquid medium which consisted of: Dextrose 20 g, yeast extract 5 g, peptone 5 g, NaCl 5 g, K_2_HPO_4_ 5 g, and distilled water to 1000 mL, with pH adjusted at 6.8 using 6N HCl [25]. Sterilization of the media was carried out by autoclaving for 20 min. at 121 °C and 15 psi.

Stage-1: Microbial cells were transferred from fresh slants (from three days to one week old) into 125 mL Erlenmeyer flasks containing 25 mL sterilized liquid medium and allowed to grow for 72 h at 27 °C on a gyratory shaker operating at 200 rpm.

Stage-2: Cultures (5 mL) of stage I culture were transferred to another 125 mL Erlenmeyer flasks containing 25 mL fresh liquid medium. Cultures were allowed to grow for 24 h before the addition of the substrate, vulgarin (5 mg/10 μL DMSO to each flask). Reaction mixtures were periodically monitored by sample withdrawal of about 1 mL of each culture (2, 6, 12, 24, 36, and 48 h of substrate incubation and every day till after two weeks of incubation). Each of the obtained samples was extracted successively with 1 mL CHCL_3_. The extracted samples were monitored on TLC silica gel 60 GF_254_ plates using 30% EtOAc in CHCl_3_ as solvent system. Visualization was accomplished by exposure to short wavelength UV (λmax 254) and spraying with vanillin sulphoric spray reagent.

Fourteen cultures showed the definite metabolism of vulgarin **(1)**.

There were: *Aspergillus niger* (ATCC 10549), *Aspergillus flavipes* (ATCC 11013), *Aspergillus versicolor* (AUMC 150201), *Cordyceps (Isaria) sinclairii* (ATCC 24400), *Cordyceps sinensis* (NBRC 610453), *Rhodotorula rubra* (NRRL y1592), *Rhizopus* species (ATCC 36060), *Mucor* species (AUMC 150209), *Gymnascella citrina* (NRRL 6050), *Alternaria alternata* (AUMC 150207), Penicillium chryzogenium (ATCC 9490), Rhizopogen sp. (ATCC 6060), and two strains of *Cunninghamella elegans* (NRRL 2310) and (NRRL 1392).

The preliminarily screening showed that *Aspergillus niger* (ATCC 10549) was the most promising microorganism as it reproducibly produced several metabolites. Therefore, a preparative scale-up fermentation study was designed to isolate, identify, and study the possible mechanism of action.

### 3.6. Large Scale Fermentation

*Aspergillus niger* (ATCC 10549) was grown in five 250 mL Erlenmeyer flasks, each containing 50 mL of liquid media. A total of 200 mg of vulgarin in 1000 μL DMSO were evenly distributed among the 24 h old stage II culture. The incubation mixture was monitored, and fermentation was terminated after 12 days by extraction with CHCl_3_ and concentration by evaporation under reduced pressure using a rotatory evaporator at 45 °C.

### 3.7. Isolation of Metabolites

The dried extract was applied on silica gel column using ethyl acetate/chloroform solvent system in gradient mode for elution analysis.

The first metabolite was eluted by 15% EtOAc/DCM, collected, pooled, and evaporated to give a pure metabolite **(2)** (20 mg). Silica gel GF_254_ TLC plate showed a single pink spot, and then a violet spot at R*_f_* = 0.57 in 50% EtOAc /DCM solvent system after heating with vanillin sulphoric spray reagent.

The epimeric mixture **(3a, b)** was eluted by 20% EtOAc /DCM. Silica gel GF_254_ TLC chromatogram showed brownish violet spot at R*_f_* = 0.46 in solvent system 50% EtOAc/DCM after heating with spray reagent. Evaporation of the collected fractions to dryness under reduced pressure yielded metabolite **(3a, b)** (8 mg).

Metabolite **(3a)** was eluted by 25% EtOAc /DCM. Silica gel GF_254_ TLC chromatogram showed a brownish violet spot at R*_f_* = 0.46 in solvent system 50% EtOAc /DCM after heating with spray reagent. Evaporation of the collected fractions to dryness under reduced pressure yielded one metabolite **(3a)** (10 mg).

### 3.8. In Vitro COX-1 and COX-2 Enzyme Inhibitory Assay

The abilities of vulgarin and its metabolites to inhibit the conversion of arachidonic acid into PGH2 were evaluated using COX-1 Cayman human enzyme inhibitory assay kit (No. 701070), COX-2 Cayman human enzyme inhibitory assay kit (No. 701080, USA), and ROBONIK P2000 EIA reader. Evaluation of the data was performed by using Four Parameter Logistic Curve online data analysis tool of MyAssays Ltd. Procedures were carried out according to manufacturer’s instructions [22,26] Celecoxib^®^ and indomethacin^®^ (Sigma-Aldrich, Burlington, MA, USA) were used as reference drugs. The selectivity indices (SI) of the tested/reference compounds (SI = IC_50_ COX-1/IC_50_ COX-2) were calculated [27].

## 4. Conclusions

Vulgarin, an eudesmanolides-type sesquiterpene lactone, was subjected to microbial transformation using several microbial strains. The most significant one was *Aspergillus niger*, as it reproducibly produced three metabolites in higher yields. The three metabolites were 1-*epi*-tetrahydrovulgarin (1α,4α-dihydroxy-5αH,6,11βH-eudesman-6,12-olide), 1α,4α-dihydroxyeudesm-2-en-5αH,6,11βH-6,12-olide, and a C-1 epimeric mixture of the second metabolite. The second metabolite and its C-1 epimeric mixture were reported here for the first time to be obtained by the biotransformation of vulgarin, as selective reduction occurred at C-1. By evaluation of the anti-inflammatory activity using the human COX inhibitory assay, vulgarin showed potent activity. The structure–activity relationship of vulgarin indicated that the C_1_-α, β unsaturated carbonyl group is essential for the activity. Additionally, the reduction of the carbonyl group at C-1 to 2ry alcoholic group either α- or β-increased the inhibitory activity toward COX-1 and decreased the inhibitory activity toward COX-2.

## Figures and Tables

**Figure 1 molecules-28-03729-f001:**
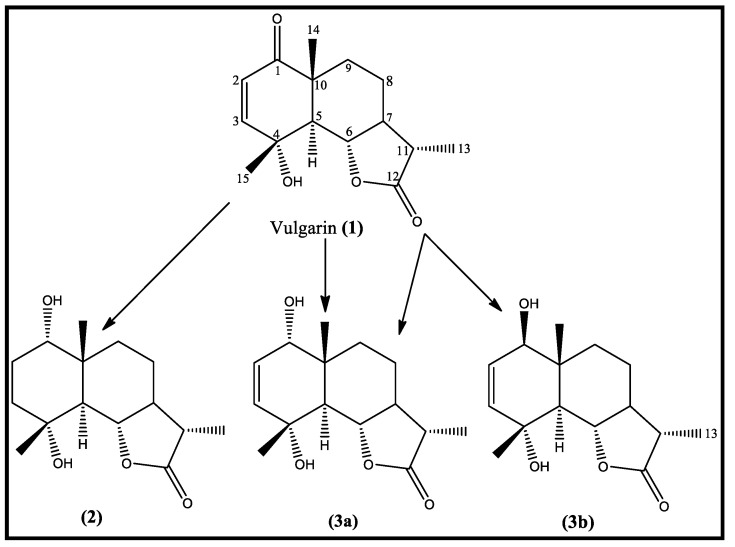
Schematic presentation of microbial transformation of vulgarin by *Aspergillus niger*.

**Figure 3 molecules-28-03729-f003:**
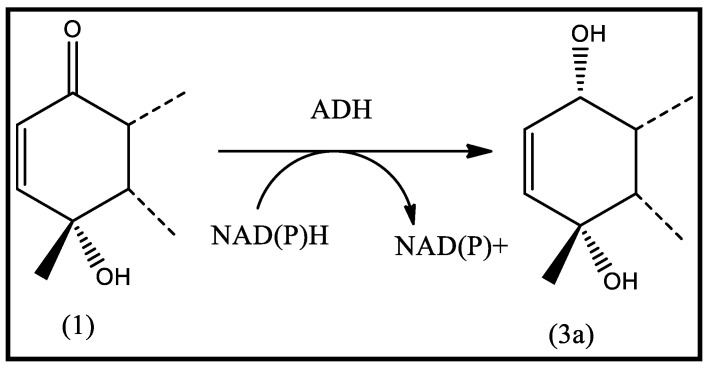
Proposed mechanism of selective reduction of C-1 carbonyl group.

**Figure 4 molecules-28-03729-f004:**
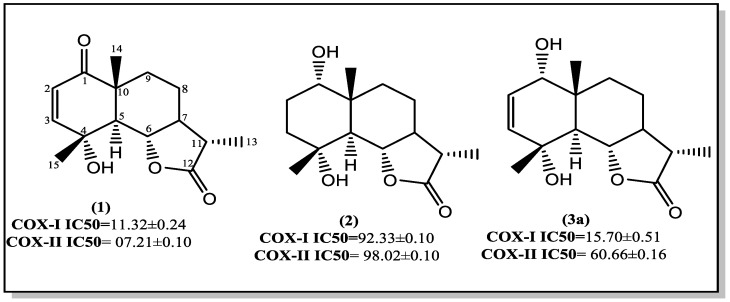
Structure of compounds with their IC50 on COX-I and COX-II enzymes.

**Table 1 molecules-28-03729-t001:** Multiplicity (*J* in parentheses in Hz) of vulgarin and its metabolites as per ^1^H-NMR spectral data in ppm.

H	1	2	3a	3b (in 3a,b)
1	-	3.41, br s	3.49, br s	3.41, br s
2	5.86, d (10.4)	1.60 m 1.91 m	5.94, dd (10.1, 5.6)	5.94, dd (10.1, 5.6)
3	6.58, d (10.4)	1.33 m 1.97 m	5.70, d (10.1)	5.70, d (10.1)
5	2.40, d (11.5)	2.18, d (11.7)	2.29, d (11.4)	2.18, d (11.6)
6	4.15, dd (10.9, 10.9)	4.07, dd (10.7, 10.6)	4.05, dd (11.4, 10.7)	4.12, dd (8.1, 4.6)
7	1.67, dddd (12.6, 12.6, 12.6, 3.5)	1.69, m	1.71, m	1.71, m
8	1.46, dddd (12.9, 12.9, 12.9, 3.2) 1.96, m	1.51, m 1.90, m	8α = 1.92, m 8β = 1.49, m	8α = 1.92, m 8β = 1.49, m
9	1.56, ddd (13.6, 13.6, 3.5) 1.99, m	1.59, m 1.95, m	9α = 2.14, m 9β = 1.32, m	9α = 1.95, m 9β = 1.25, m
11	2.34, dq (13.7, 6.9)	2.27, dq (12.6, 7.0)	2.33, dq (12.5, 6.9)	2.33, dq (12.5, 6.9)
13	1.22, d (6.9)	1.23, d (6.9)	1.25, d (6.9)	1.22, d (6.9)
14	1.19, s	1.00, s	0.97, s	1.00, s
15	1.53, s	1.36, s	1.39, s	1.36, s

**Table 2 molecules-28-03729-t002:** Vulgarin and its metabolites’ ^13^C-NMR spectral data.

C	1	2	3a	3b (in 3a,b)
1	202.1 C	73.3 CH	71.3 CH	73.4 CH
2	125.9 CH	33.0 CH_2_	127.3 CH	127.3 CH
3	152.2 CH	36.4 CH_2_	136.2 CH	136.2 CH
4	70.4 C	71.5 C	70.4 C	71.3 C
5	54.9 CH	52.9 CH	48.9 CH	50.7 CH
6	79.9 CH	81.4 CH	81.3 CH	81.5 CH
7	52.7 CH	50.5 CH	52.5 CH	53.0 CH
8	23.0 CH_2_	23.2 CH_2_	23.2 CH_2_	23.2 CH_2_
9	34.6 CH_2_	26.6 CH_2_	35.5 CH_2_	36.5 CH_2_
10	46.6 C	41.6 C	40.3 C	41.7 C
11	40.9 CH	40.7 CH	40.9 CH	41.7 CH
12	178.7 C	178.7 C	179.0 C	179.0 C
13	12.8 CH_3_	12.5 CH_3_	12.6 CH_3_	12.6 CH_3_
14	20.1 CH_3_	20.0 CH_3_	20.2 CH_3_	20.1 CH_3_
15	24.1 CH_3_	24.3 CH_3_	24.6 CH_3_	24.4 CH_3_

**Table 3 molecules-28-03729-t003:** Percentage of inhibition of COX-1 and COX-2 of tested compounds.

Compounds	COX-1%	COX-2%
Vulgarin **(1)**	45.61%	60.69%
**2**	9.43%	6.80%
**3a**	53.60%	22.7%
**3a,b** (epimeric mixture)	48.40%	26.17%
Indomethacin	85.68	14.89%
Celecoxib	8.7%	92%

**Table 4 molecules-28-03729-t004:** IC_50_ of Tested compounds on COX-1 and COX-2.

Compounds	COX-1 IC_50_ * (μM/mL)	COX-2 IC_50_ * (μM/mL)	Selectivity Index: (IC_50_ COX-1/ IC_50_COX-2)	Selectivity Index: (IC_50_ COX-2/IC_50_ COX-1)
Vulgarin **(1)**	11.32 ± 0.24	07.21 ± 0.10	18.53	0.64
**2**	92.33 ± 0.10	98.02 ± 0.10	0.94	1.06
**3a**	15.70 ± 0.51	60.66 ± 0.16	0.26	3.86
**3a,b**	10.12 ± 0.01	64.22 ± 1.05	0.16	6.35
Indomethacin	0.24 ± 0.05	3.28 ± 0.09	0.07	13.67
Celecoxib	29.19 ± 0.33	0.08 ± 0.44	364.88	4.56

* All data are represented as mean value ± SD for three independent experiments.

## Data Availability

Not applicable.

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
