# Peer review of "Microbial Transformation of the Sesquiterpene Lactone, Vulgarin, by Aspergillus niger"

_molecules, 2023, doi:10.3390/molecules28093729_

Round 1

Reviewer 1 Report

In this manuscript, the authors investigated the microbial transformation of an eudesmanolides-type sesquiterpene lactone, and three metabolites were isolated and characterized. The anti-inflammatory activity of Vulgarin and its metabolites were also evaluated. Minor revision is recommended, and the comments are as below:

1.     The previous reports of Vulgarin metabolites should be discussed in the “Introduction”.

2.     How many replicates of the inhibitory assay?

3.     The integration of the signal in 1H NMR spectrum need be modified.

4.     The 13C-NMR spectrum of (3a) and (3a,b) needs to be improved, the peak signal is quite weak and difficult to distinguish from background noise.

5.     Any proposed mechanism for the generation of Vulgarin metabolites.

6.     The author should carefully check the format of text, there should be a space between the number and the degree symbol, such as “Dextrose 20g” in line 114.

7.     Abbreviations appeared the first time should be described in full name, such as “COX”.

8.     In “abstract” line 21, “activi ty” should be “activity”?

Author Response

Reviewer 1:

Reviewer comment

Authors response

1.  1. The previous reports of vulgarin metabolites should be discussed in the “Introduction”.

Thank you so much. We have added the previous reports of vulgarin metabolites in the fourth paragraph in the introduction.

2.     2. How many replicates of the inhibitory assay?

We appreciate your question.

The answer is Triplicate.

*All data are represented as mean value ± SD for three independent experiments.

This sentence is added below Table 4.

3.     3. The integration of the signal in 1H NMR spectrum need be modified.

Thank you for your comment.

The integration of  1HNMR  is automatically achieved into figures S1, S8, S13, and S20.

4.   4. The 13C-NMR spectrum of (3a) and (3a,b) needs to be improved, the peak signal is quite weak and difficult to distinguish from background noise.

Thanks for the comment.

Despite a high number of scans, the 13CNMR signals are not sharp only for a few quaternary carbon peaks, which may be due to the low yield of both compounds.  The clear contours in 2D HSQC and HMBC spectra were used to assign each proton and carbon signal.

5.  5.  Any proposed mechanism for the generation of Vulgarin metabolites.

We appreciate your comment.

Proposed mechanism of selective reduction is added under discussion with a figure.

Section 3.2.

6.  6. The author should carefully check the format of text, there should be a space between the number and the degree symbol, such as “Dextrose 20g” in line 114.

Thank you for your note. Correction is done.

7.     7- Abbreviations appeared the first time should be described in full name, such as “COX”.

Thank you so much.

We wrote the full name (cyclooxygenase) before the abbreviation, COX.

8.     8- In “abstract” line 21, “activi ty” should be “activity”?

Thanks for your note.

Correction is done.

Reviewer 2 Report

This present manuscript was aimed to utilize microorganisms to investigate the biotransformation of vulgarin for the production of metabolites with enhanced anti-inflammatory activity. The structures of vulgarin (1) and its metabolites (2), (3a), and (3b) were elucidated by 1D & 2D NMR spectroscopy in conjunction with HRESIMS. Metabolites (3a) and (3b) are epimers and reported here for the first time as new metabolites obtained by biotransformation through selective reduction at C-1. Although I appreciated authors′ efforts, the present manuscript still displayed various errors including the typography and experimental design. The present results only displayed preliminary advances and no significant improvements on the bioactive components discovery related fields could be observed. In summary, this manuscript is not recommended to accept for publication in Molecules. Moreover, the following addressed comments should be carefully revised before the consideration of its resubmission.  

1.         There were still some typographic, grammar, and format errors to be observed in the text. Authors have to check and revise these errors carefully.

2.         The aim of this study was to discover metabolites with enhanced anti-inflammatory activity, however, the resulted constituents did not show improved bioactive data. In addition, authors did not perform the study according to the bioassay guided purification procedures.

3.         Authors did not purify (3b) and only assigned its spectral characteristics in the mixture. It is not completely allowed in natural products chemistry research since 3a and 3b should be separated well by HPLC.

4.         In Materials and Methods section, authors should provide the complete spectroscopic and spectrometric data of the isolated compounds.

5.         The screening results of different microorganisms should be briefly mentioned in the text.

6.         The structural elucidation section should be separated into some paragraphs to make it clear for the readers. Some 2D results should be noted in the figures. This section could be further improved with the assistance of editing by some professional experts.

7.         The bioactivity data of these metabolites were not significant as compared with the precursor.

Author Response

Reviewer 2:

Reviewer comment

Authors response

1.     1.  There were still some typographic, grammar, and format errors to be observed in the text. Authors have to check and revise these errors carefully.

We appreciate your notes.

The manuscript was revised twice.

2.     2. The aim of this study was to discover metabolites with enhanced anti-inflammatory activity; however, the resulted constituents did not show improved bioactive data. In addition, authors did not perform the study according to the bioassay guided purification procedures.

Thank you so much for your comments.

The authors hoped to discover more active metabolites. However, in this study, we also aimed to:

1.      Study the structure-activity relationship by comparing the biological activity of compounds of closely related structures.

   2. Study the metabolism of vulgarin by the liver

       enzymes when used as an anti-inflammatory drug

       depending on the capability of microbial

       transformation to mimic mammalian metabolism.

    3.  Moreover, this work aimed to study enzymatic

         reactions carried out by microorganisms which

         have the advantage of selectivity over chemical

         reactions (These goals were added at the end of the introduction).

In microbial transformation we extracted vulgarin and its metabolites from the culture media with chloroform, isolated the metabolites with different chromatographic columns and finally perform the biological assay.

3. Authors did not purify (3b) and only assigned its spectral characteristics in the mixture. It is not completely allowed in natural products chemistry research since 3a and 3b should be separated well by HPLC.

 We appreciate your question.

 The amount that remained after purification was very minimal following numerous attempts to separate 3a from 3b using various chromatographic techniques, including open column and preparative TLC using different adsorbents, such as normal and reversed-phase silica gel. Additionally, we lack an axis for preparative HPLC. Some references, including the one that is attached, identified the isomers in a mixture as shown below.

Hassan AR, Ashour A, Amen Y, Nagata M, El-Toumy SA, Shimizu K. A new cycloartane triterpene and other phytoconstituents from the aerial parts of Euphorbia dendroides. Nat Prod Res. 2022 Feb;36(3):828-836. 

4.     4. In Materials and Methods section, authors should provide the complete spectroscopic and spectrometric data of the isolated compounds.

Thank you for your comment.

In the material and method section, the isolation of metabolites was written. While structural elucidation with spectroscopic and spectrometric data of the isolated compounds were written under discussion.

5.  The screening results of different microorganisms should be briefly mentioned in the text.

Thank you for your comment.

We carried out screening using 20 microorganisms. They should be written in detail with their codes to be accurate in determining the strains used.  

.           6. The structural elucidation section should be separated into some paragraphs to make it clear for the readers. Some 2D results should be noted in the figures. This section could be further improved with the assistance of editing by some professional experts.

Thank you so much for your comment.

The structural elucidation section is separated into sections and some paragraphs

3.1.1. Structural elucidation of metabolite (2)

3.1.2. Structural elucidation of metabolite (3a)

3.1.3. Structural elucidation of metabolite (3a,b)

A figure for some  2D results is added (Figure 2)

7.          7. The bioactivity data of these metabolites were not significant as compared with the precursor.

We appreciate your effort.  This point discussed in 2

Round 2

Reviewer 2 Report

I had gone through the revised manuscript and my comments were attached herein. Although authors had provided some responses as compared with the previous version, the key point of “new” metabolite 3b was still the major concern of this manuscript. Authors cited a reference to support their manuscript, however, it was the different situation. Both epimers were new in the reference, in contrast, the only new compound in this study was a mixture. If authors publish it as a “new compound”, other publications later could only report it as a “known compound”. It is not fair for other researchers. In addition to this point, I felt that the present manuscript may meet the minimum criteria of Molecules and this manuscript could be reconsidered for its publication after major revision described above.

Author Response

Dear Prof

We added some sentences to preserve the right if somebody in the future isolated the epimers in pure isomeric form. Also the manuscript described other finding rather than getting new compounds. 

Reviewer comment

Authors response

I had gone through the revised manuscript and my comments were attached herein. Although authors had provided some responses as compared with the previous version, the key point of “new” metabolite 3b was still the major concern of this manuscript. Authors cited a reference to support their manuscript, however, it was the different situation. Both epimers were new in the reference, in contrast, the only new compound in this study was a mixture. If authors publish it as a “new compound”, other publications later could only report it as a “known compound”. It is not fair for other researchers. In addition to this point, I felt that the present manuscript may meet the minimum criteria of Molecules and this manuscript could be reconsidered for its publication after major revision described above.

 We appreciate your comments.

The amount of epimeric mixture separated from microbial transformation was 8 mg. Apart of this amount was consumed in several attempts in bioassay and in spectral and mass analysis and the remaining amount is about 4 mg. Furthermore, compound 3b is minor in the mixture as its ratio is 1:4 (3b/3a), so its amount is about 1mg in the mixture.     

 We tried recently to separate them but the remaining amount is very small. However, we tried to separate them by preparative normal or reversed silica gel TLC, but the 2 epimers always appear as one spot which make it very difficult to be separated. If we success to separate them, we will face another problem as we will not be able to carry out spectral analysis for a compound of 1 mg only. In addition the manuscript is not based only on getting new compounds alone but also on biotransformation methods and biological activities of the obtained metabolites.
